# Self-Attention for Quantum Entanglement Prediction

**Anuj Gore**
Department of Physics and Astronomy,
University College London, WC1E 6BJ, U.K.
`anuj.gore.22@ucl.ac.uk`

**Dylan Lewis**
Blackett Laboratory, Imperial College London,
London, SW7 2AZ, U.K.

**Roopayan Ghosh**
School of Basic Sciences,
Indian Institute of Technology,
Bhubaneswar, 752050, India

**Sougato Bose**
Department of Physics and Astronomy,
University College London,
London, WC1E 6BJ, U.K.

## Abstract

Quantum entanglement is a powerful resource in quantum mechanics and quantum information processing. However, its reliable quantification remains challenging due to the exponential growth of the underlying Hilbert space with system size, which renders full state reconstruction infeasible. Moreover, experimentally estimating entanglement typically requires a large number of measurement samples leading to a significant overhead. In this paper, we present two models, a feed-forward neural network and an attention-based model, to accurately predict the entanglement of random states. Our results demonstrate that machine-learning method consistently outperform conventional analytical approaches across a range of qubit numbers, highlighting the advantages of machine learning for the efficient quantification of quantum resources. Our work is publicly available at `https://github.com/AnujGore/renyi2-prediction-ml`

## 1 Introduction and Motivation

In recent years, quantum computing has emerged with the potential to transform industries by solving complex problems beyond the capabilities of classical computers by leveraging key quantum physics concepts such as superposition and entanglement. Quantum entanglement is widely used in quantum communication via teleportation (Bouwmeester et al., 1997), speed-up in quantum computation (Jozsa & Linden, 2003), is essential in quantum cryptography (Yin et al., 2017) and more. Due to the inherent complexity and the scaling of the space an arbitrary quantum state (or system) lies in, it is known to be NP-hard to quantify entanglement (Schneeloch & Howland, 2018). A robust approach to quantifying entanglement is using the technique of classical shadows (Huang et al., 2020). This method has found itself widely adopted in modern calculations such as in quantum error mitigation (Cai et al., 2023), quantum machine learning (Huang et al., 2021), and other many-body problems (Huang et al., 2022a).

Machine learning has experienced rapid growth in recent years, driven by advancements in computational power, data availability, and algorithmic innovation, with the arrival of the Transformer model (Vaswani et al., 2023). In the context of how machine learning can aid quantum computation, techniques such as reinforcement learning can optimize quantum control and guide experiments, making quantum systems more stable and scalable (Havlíček et al., 2019). Genetic algorithms have shown to aid quantum annealing in reaching the final state solution with a high probability (Hegde et al., 2022). Recurrent networks have seen themselves useful in decoding post error corrected bits (Bausch et al., 2024).

In our work, we advance the ability of a machine learning model to aid quantum systems by comparing a self-attention model and a feed forward model with the inputs of the classical shadows of a Haar random quantum system to predict the Rényi-2 entropy. While attention mechanisms have been explored in quantum settings, including circuit partitioning (Russo et al., 2024), quantum full

state tomography (Cha et al., 2021), wave-function reconstruction (Zhang & Di Ventra, 2023), and the simulation of decohering systems (Luo et al., 2022), their application to entanglement prediction remains limited. Prior machine learning approaches to entanglement have focused on detecting the presence of entanglement (Asif et al., 2023; Ureña et al., 2024), estimating specific measures such as logarithmic negativity using system moments (Gray et al., 2018), or inferring entanglement from local expectation values (Huang et al., 2022b). Therefore, to the best of our knowledge, we introduce a novel framework that enables experimental quantum computing platforms to estimate the entanglement of Haar-random quantum states using only computational-basis measurements, without requiring full quantum state tomography or explicit wave-function reconstruction. Our approach is scalable, permutationally invariant, and naturally generalizes to larger qubit systems, making it suitable for near-term and large-scale quantum devices.

## 2 BACKGROUND

With the formal definitions summarized in Appendix A, we consider quantum systems defined on a tensor-product Hilbert space $\mathcal{H} = \mathcal{H}_A \otimes \mathcal{H}_B$. In this setting, entanglement refers to the non-separability of the state with respect to this tensor-product structure, i.e., the impossibility of writing it as a product (or convex mixture of products) of states on the subsystems (Plenio & Virmani, 2006). In this work, we focus on pure states, for which entanglement is fully characterized by the spectrum of the reduced density matrix. Accordingly, we quantify entanglement using the second Rényi entropy, defined as

$$H_2(\rho_A) = -\log_2 \text{Tr}(\rho_A^2) \tag{1}$$

where, $\rho_A$ is the reduced density matrix for subsystem $A$ of the normalized density matrix of the state in question, $\rho$. It is known that full state reconstruction of an arbitrary quantum system is expensive in the number of samples, complexity and memory. A feasible alternative is the classical shadow protocol, where, through rotating a system with a given unitary and sampling, we can estimate its quantum properties without requiring full state tomography, giving a complexity $\mathcal{O}(d_A^2)$ in the subsystem size $d_A = \dim(\mathcal{H}_A)$ (Huang et al., 2020). In fact, Brydges et al. (2019) showed that the Rényi entropy, a quantum property, can be computed from the same ingredients - randomized unitaries and measurement outcomes. The entropy can be found from

$$H_2(\rho_A) = -\log_2 \bar{X} \tag{2}$$

$$X = 2^{N_A} \sum_{s_i, s_j} (-2)^{D[s_i, s_j]} P(s_i) P(s_j), \tag{3}$$

where $N_A$ indicates the number of qubits we perform the partition over, and the sum over $s_i, s_j$ indicates a summation over all logical basis states. Each basis state is given as a bit string in $\{0, 1\}^{\otimes N}$ for $N$ qubits. $D[s_i, s_j]$ is the Hamming distance between the strings of the logical basis states and $P(s_q)$ is the probability of observing state $s_q$, driven by $\langle s_q| U_i \rho U_i^\dagger |s_q\rangle$ for all basis states given by $s_q$. Each unitary, $U_i$ is sampled from the Haar measure on $N$ qubits (Lewis et al., 2025),

$$U = \exp\left(-i \sum_{j=1}^{4^N - 1} \theta_j P_j\right) \tag{4}$$

where, $\theta_j \sim [\frac{\pi}{2}, \frac{\pi}{2})$ and $P_j \in \mathcal{P}(N_A)$ where $\mathcal{P}(N_A)$ is the Pauli group on $N_A$ qubits defined as:

$$\mathcal{P}(t) = \{\sigma_1 \otimes \cdots \otimes \sigma_t \mid \sigma_i \in \{\mathbf{I}, \mathbf{X}, \mathbf{Y}, \mathbf{Z}\}, \ i = 1, \ldots, t\} \setminus \{\mathbf{I}^{\otimes t}\}.$$

where, $\mathbf{I}, \mathbf{X}, \mathbf{Y}, \mathbf{Z}$ are the Pauli matrices for 2 qubits. $\mathcal{P}(t)$ has set cardinality of $4^N - 1$.

To determine $P(s_q)$, we need to perform multiple measurements of the system ($N_M$) as the quantum system returns a probabilistic outcome. By rotating the system with multiple different unitaries, we can capture more of the geometry of the state. The bar (e.g. $\bar{X}$) denotes the ensemble average over $N_U$ unitaries. This set of $N_U$ unitaries, with each unitary being measured $N_M$ times, is passed as input to our models. As we have numerical access to the full quantum system, it is trivial to

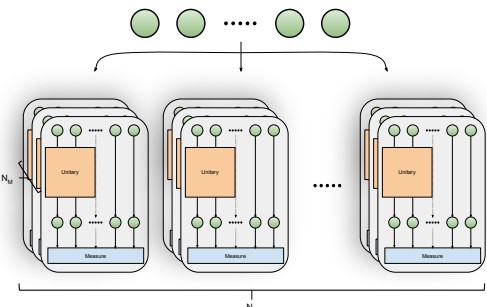

Figure 1: Schematic workflow of our data generation model. Note that the green spheres represent qubits and we consider a Haar random state during data generation.

calculate our target value of the Rényi-2 entropy. This data generation flow is illustrated in Fig 1 with additional details in Appendix B.

To ensure resource efficiency, we bound the number of unitaries and measurements (or shots) needed to successfully estimate the whole geometry of the $N$ qubit system. Our calculations on these bounds are presented in Appendix C. We aim to reduce both $N_U$ and $N_M$ via machine learning techniques, thereby improving the efficiency of quantum state characterization.

## 3 MODEL

In this section, we detail the architecture of the model. We employ two networks for our entanglement quantification: a feed-forward network (MLP), and a self-attention model. Both models process the inputs in two stages. In the first stage, a feed-forward neural network, $\phi$, is implemented to transform our unitary (which is in $\mathbb{C}^{2^N \times 2^N}$) to a vector in $\mathbb{R}^d$. The value of $d$ chosen is $2^N$, where $N$ is the number of qubits. The value of $d = 2^N$ worked well in practice but could be any value. Optimising on this parameter choice was not necessary for our results, but could be used to improve results for larger qubit systems. It is worth noting that this feed-forward network extracts features that the model deems relevant towards quantifying the Rényi entropy of the system, as the gradient values are dependent on the loss function between the predicted entropy and true entropy. As a result, this transformed unitary no longer retains all the information of the unitary. It contains the extracted features from our initial unitary with respect to the entropy of a system. In our second stage, we concatenate the outcome probabilities to this transformed unitary and pass it through a final neural network $S$, which is used to predict the entanglement. For the self-attention model, between these two stages, we have a self-attention step further detailed in Section 3.2.

The pre-processing of our data was to count the total number of occurrences of each basis state over $N_M$ measurements and then computing the average for each basis state; this reflected the probability of a certain basis state occurring. Therefore, as input to our model, we have $N_U$ probability arrays (averaged over $N_M$ measurements), each in $\mathbb{R}^{2^N}$ and $N_U$ unitaries, each in $\mathbb{C}^{2^N \times 2^N}$.

### 3.1 ARCHITECTURE OF MODELS

Since a linear neural network processes one-dimensional input vectors, we represent each unitary as a vector in operator Hilbert space via column-wise vectorization (Choi isomorphism), to form $N_U$ vectorized unitaries in $\mathbb{C}^{4^N}$. We show the structure of both our models in Fig. 2.

Our second model employs self-attention. The formula for attention is given as,

$$\text{Attention}(Q, K, V) = \text{softmax}\left(\frac{QK^T}{\sqrt{d_k}}\right) V \tag{5}$$

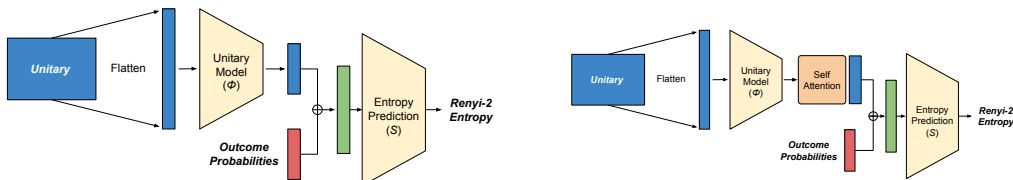

Figure 2: *Left:* MLP model design. *Right:* MLP with Self-Attention - we implement the self-attention after the unitary encoding but before the prediction. Note that the bold and italicized terms are our input and output. The transformed unitary and the size of our outcome probabilities are identical.

and in case of self-attention, $Q = K = V = U'$. Recall that $U'$ is the set of transformed unitaries after applying $\phi$. During inference, the attention weights $(QK^T)$ remain frozen and we compute the direct matrix-vector multiplication between them and our transformed unseen unitaries. As observed from Equation equation 5, there are no extra parameters, admitting that the feed-forward network and the self-attention employed model have the same size and parameter count. Despite introducing no additional trainable parameters, the self-attention layer plays a crucial role in the generalizability of the model to other forms of entanglement.

## 3.2 CONSTRUCTION OF ATTENTION WEIGHTS

We present our method of calculating the attention matrix. Recall the expression for sampling a unitary $U$, Equation equation 4. We perform the column-wise vectorization of $U$, where each matrix element of $U$ can be indexed as $U_{i,j}$. During vectorization, we explicitly keep track of the mapping between the matrix indices $(i, j)$ and the resulting vector index resulting in a vectorized unitary, $\vec{u}^{(s)}$, where the additional label of $s$ denotes the $s$-th unitary sampled,

$$\vec{u}^{(s)} = \sum_{i,j=1}^{2^N} U_{i,j} \, |i,j\rangle \, . \tag{6}$$

To form our attention weights, we then calculate our outer product as,

$$\mathcal{A} = \sum_{s=1}^{N_U} |\vec{u}^{(s)}\rangle \langle \vec{u}^{(s)}| \tag{7}$$

for $N_U$ sampled unitaries. This outer product induces a Gram matrix over the sampled unitaries, encoding their pairwise overlaps in the $N$-qubit operator Hilbert space. Such Gram matrices are widely used to characterize geometric and statistical structure in high-dimensional feature spaces:

- **Projector onto the operator subspace.** The support of $\mathcal{A}$ coincides with the linear span of the ensemble $\{\vec{u}^{(i)}\}_{i=1}^{K}$, for $K$ unitaries. Then, up to normalization, $\mathcal{A}$ acts as a projector onto the space of the unitaries, making it a natural object for subspace identification.

- **Beyond pure state constructions.** $\mathcal{A}$ encodes the ensemble structure of random unitaries. Following Schrodinger-HJW theorem (Hughston et al., 1993), that states that every mixed state can be represented as an ensemble of pure states decomposed corresponding to a set of unitaries on the support of $\mathcal{A}$. Thus, this Gram matrix provides a natural structure for reconstructing or characterizing all valid pure-state decompositions of a given mixed state.

This formalism also lends itself in understanding the geometric measures of entanglement as developed by Chen et al. (2011) and Gielerak & Sawerwain (2019). In our case, we transform each unitary via $\phi$ to $\mathbb{R}^{2^N}$, thereby eliminating the explicit matrix index structure. As a result, not only is our Gram matrix of our transformed unitaries, but it also contains features relevant to the quantification of the entropy of the system as, during back-propagation, the gradients of $\phi$ are trained with respect to the Rényi entropy.

As stated in Equation equation 2, the Rényi entropy of a system can be found from the unitaries and the outcomes. Specifically, the unitaries and the outcomes determine the purity of the system, from which the quantification of the Rényi entropy is trivial. From the purity, the linear entropy (purity - 1), negativity ($\sqrt{(\text{purity} - 1)/2}$), and local properties of the state (Inverse Participation Ratio) can be calculated. Purity has also been shown to provide a measure of multi-qubit entanglement that is a function on pure states (Brennen, 2003). As a result, we posit that this Gram matrix could be suited for entropy and purity based calculations without reconstructing $N_U$ unitaries. In this work, we focus on the Rényi entropy.

## 4 RESULTS

The simulation of the quantum system is written in PyTorch, developed and managed by Meta (Paszke et al., 2019). The models are written in Jax (Bradbury et al., 2018) and Flax (Heek et al., 2024), both developed and managed by Google. We generated 10000 Haar random states with $(N_U, N_M)$ equal to $(100, 200)$ for 2 qubits and $(500, 200)$ for 4 qubits. 70% of this generated set is used for training and 30% is used for evaluation, which occurs after every 10 epochs. The training lasts for 5000 epochs. Using logarithmic spacing for $N_M$ and linear spacing for $N_U$, we generated an additional 1000 copies for evaluation on different $(N_U, N_M)$ tuples. Other implementation details such as hyperparameters, hardware details, parameter count are in Appendix D.

### 4.1 2 QUBIT RESULTS

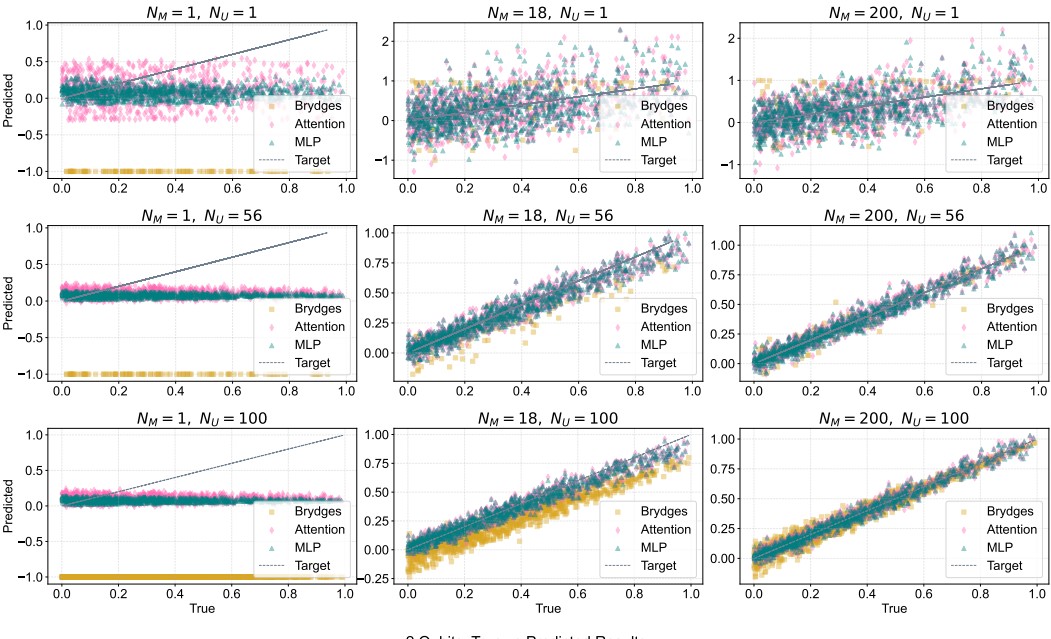

Figure 3: Correlation between True and Predicted entropy for 2 qubits for different $N_U N_M$ pairs using the attention model and the MLP against the analytical solution. The dotted line represents the ideal spread of the target values; the golden dots represent the analytical solution, blue dots for our MLP and hotpink for the self-attention model.

In Fig 3 we plot the computed entropy against the true entropy for all three cases - the self-attention model, MLP and Brydges - to inspect the behaviour of the models for highly entangled and highly separable states. We plot 9 ($3 \times 3$) inset plots with increasing measurements per unitary from left to right and increasing unitaries from top to bottom. Regardless of the model choice, following logic, we observe a better prediction of the entropy for a higher number of $N_U$ and $N_M$. Increasing the number of unitaries for a low number of measurements, the analytical solution concentrates around

$-1.00$ as a result from the $-2$ factor in Equation 2. However, the learning models are distributed around 0, with a sparse behaviour for a lower $N_U$ and becoming concentrated with increasing $N_U$. This could be due to the LeCun initialization of the weights (LeCun et al., 1998), which centre around 0 and have a variance depending on the dimension of the layer. Repeated activation of these weights, with no learning, compounds the value of the weights resulting in the concentration seen as $N_U$ increases. Increasing $N_M$, independently of $N_U$, results in the model learning some information about the state. In all model cases, we see that the general slope of the predictions start to resemble ideal predictions.

In analysis of our models, comparing the machine learning models to the analytical solutions, we see a lower error in the highly separable states (bottom left corner of subplots) and a similar error in highly entangled states (top right corner of subplots). We attribute the reasons for the model's tighter bounds on the error in the highly separable states to the abundance of the training samples available for a highly separable states. Conversely, in the highly entangled region, as the analytical formula is deterministic, having the machine learning model exhibit similar variance is evident that the model learns the underlying structure of the system. To further inspect this variance and for a rigorous analysis of the models, we plot a heat-map of the error from the true value with increasing $N_M$ from bottom to top and increasing $N_U$ from left to right and highlight points with error lower than a certain value including the variance in Fig 4 for all three models.

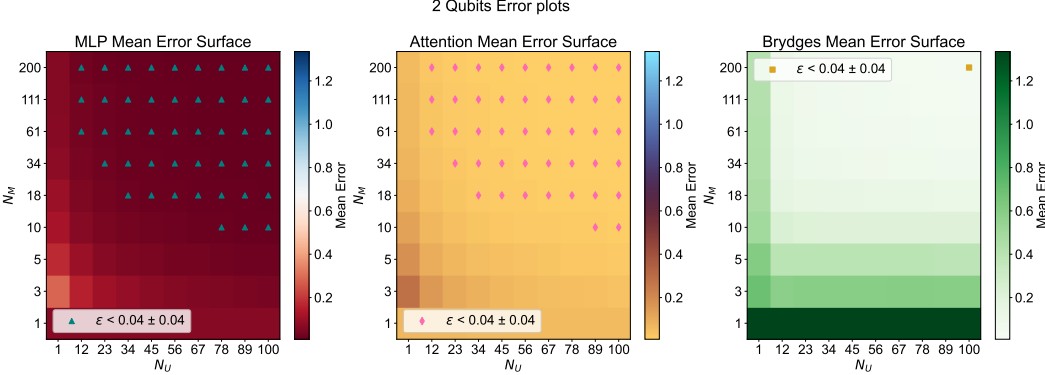

Figure 4: Accuracy of entropy prediction for 2 qubits for varying $N_U$ and $N_M$ ranges. In the plots, we calculate the mean squared error of each method (sub-labels) in predicting the entropy of a Haar random system. The highlighted points represent the $(N_U, N_M)$ tuples that have the error *and* uncertainty of error below the value in the legend ($\varepsilon$).

In Fig 4, the highlighted points denote the $(N_U, N_M)$ tuples that are on average within the error and variance of $0.04 \pm 0.04$. This figure further inspects the quality of the analytical formula by constraining the variance in the error. This figure also demonstrates the strength of the machine learning based methods as we see a larger number of highlighted points compared to the analytical formula, indicating that the model predicts the entropy within the variance limit accurately in a lower number of unitaries and measurements compared to the analytical formula. Comparing the MLP model against the self-attention model, for a low number of measurements, we see that the MLP model can estimate the entropy in less than 80 unitaries compared to the attention model needing minimum 89. We attribute this to the MLP processing the inputs and learning the brute-force solution for the entropy.

## 4.2   4 QUBIT RESULTS

Fig. 5 has the same structure as that of the Fig 3, with the exception of the number of unitaries to 500 for 4 qubits. Similar to the result of 2 qubits, as the $(N_U, N_M)$ tuples increase in value, our models converge to the solution more accurately. By increasing the number of unitaries for a low number of measurements, similar to 2 qubits, our data points seem to concentrate around slightly above 0, reiterating our argument of the compounded LeCun initialized weights converging to a mean value of 0. Additionally, from the plots we observe two clearly distinct effects associated with $N_M$ and $N_U$.

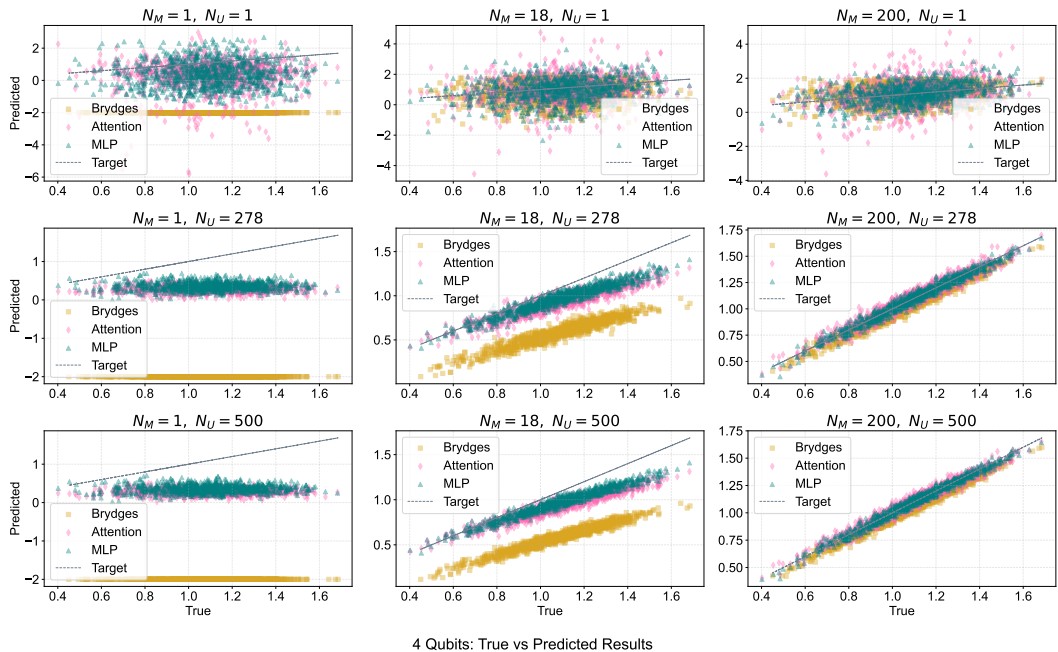

4 Qubits: True vs Predicted Results

Figure 5: Correlation between True and Predicted entropy for 4 qubits for different $N_U N_M$ pairs using the attention model and the MLP against the analytical solution

Increasing the number of measurement shots $N_M$ primarily reduces the variance of the predicted entropy values around the true entropy. In other words, larger $N_M$ suppresses statistical fluctuations arising from finite sampling and improves the precision of the estimator without significantly altering its overall trend. In contrast, increasing the number of random unitaries $N_U$ predominantly corrects systematic deviations in the slope of the predicted versus true entropy curve. In particular, larger $N_U$ improves agreement in the high-entanglement regime, bringing the predicted values into closer alignment with the ideal entropy line. This behavior reflects the distinct roles of the two resources: $N_M$ controls statistical shot noise for a fixed unitary, whereas $N_U$ improves the approximation to the unitary ensemble average and therefore reduces systematic bias in the entropy estimation.

We now turn to the analysis of the three models for four qubits. We find that the machine learning estimators exhibit a lower variance in the error than the analytically derived estimator when applied to finite-sample measurement data. As the sampling method limited the number of highly separable and highly entangled state generation, and yet the models accurately predicted the entanglement, we can say with confidence that the machine learning models understood the underlying geometry and offer more robust and reliable solution. To further inspect the models for a rigorous analysis, similar to Fig 4, we use a heat-map to visualize the error of the models for different $(N_U, N_M)$ tuples in Fig 6.

In Fig 6, the highlighted points denote the $(N_U, N_M)$ tuples that are on average within the error and variance of $0.06 \pm 0.1$. By constraining the variance, we see that the machine learning models (in general) can predict the entropy on a lower number of measurements. While the analytical solution requires a lower number of unitaries than our self-attention model, the MLP can predict the entanglement in more than 50 fewer unitaries than the analytical model. Comparing the MLP model against the attention model, we see that the MLP model can estimate the entropy in fewer shots and fewer unitaries which we once again attribute to the MLP processing the inputs and learning the brute-force solution for the entropy.

Overall, our results show that the machine learning based models reduce the number of unitaries and significantly reduce the number of measurements (per unitary) needed to compute the Renyi-2 entropy of a pure state system. We attribute this to the ability of the model to learn the underlying mechanisms of the quantum system and interpret the state geometry to accurately predict the entropy.

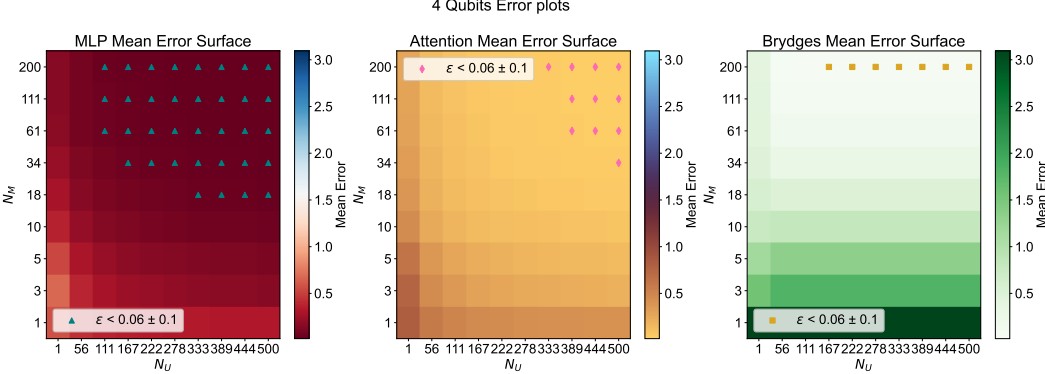

Figure 6: Accuracy of entropy prediction for 4 qubits for varying $N_U$ and $N_M$ ranges.

For both 2 and 4 qubit systems, the self-attention model and the MLP increasingly approach ideal predictions as the number of sampled unitaries ($N_U$) and measurements per unitary ($N_M$) grow, with the MLP consistently achieving accurate estimates using fewer resources. Although Haar random sampling produces a Page-like distribution of target entropy values (see Appendix B), the learning models are able to infer the underlying structure of the system and generalize across different ($N_U, N_M$) tuples. Error heat-maps reveal that the learned models exhibit tighter variance bounds and substantially reduced dependence on measurement count compared to the analytical solution, particularly in low-shot regimes, becoming a useful tool in error-prone measurements currently seen on Noisy-Intermediate Scale Quantum devices.

In Appendix E, Fig 11, we plot additional figures from the ones above. Namely, we extrude the heat-maps by placing the error on the z-axis, which gives us a better view of how the models infer the classical shadows for different ($N_U, N_M$) tuples. Additionally, in Fig 10, we use the scatter plots from Fig 3 and Fig 5 to estimate the line of best fit, whose slope and intercept we plot. This figure gives us a better view of the convergence of the model and how the selection of the number of measurements and unitaries can affect the result.

## 5   OUTLOOK

In this work, we have introduced a machine-learning-assisted framework for estimating the second order Rényi entropy of Haar-random quantum states from randomized unitary measurements, achieving a substantial reduction in the number of measurement shots required relative to analytically derived estimators. By learning directly from finite-sample data, our approach remains accurate and stable in regimes where traditional randomized-measurement formulas become noise-dominated, and this advantage persists as the system size increases. This makes the method particularly promising for scalable entanglement characterization on near-term quantum devices.

By benchmarking against the protocol of Brydges et al. (2019), we demonstrated that reliable Rényi-2 entropy estimates can be obtained using far fewer random unitaries and significantly fewer measurement shots than would be required by analytical estimators applied to the same finite data. Remarkably, this performance persists despite the strong Haar-induced bias toward near-maximal entanglement, indicating that the learned models capture non-trivial geometric structure in the space of quantum states rather than reproducing typical values. This establishes data-driven inference as a powerful route toward low-shot entanglement estimation in experimentally realistic settings.

Our results also highlight both the promise and the present limitations of this approach. Because Haar-random states are strongly biased toward near-maximal entanglement, weakly populated regions of the entanglement spectrum - which are relevant for fault-tolerant protocols - remain statistically under-represented. Nevertheless, these weakly entangled states are generated on near-term quantum devices. Followed by this preparation and distillation, highly entangled resource states can be produced, placing our method squarely within the most experimentally relevant regime. Haar

simulation becomes exponentially costly in classical memory beyond modest qubit numbers and alternative physically motivated ensembles - such as transverse-field Ising, XY, and tensor-network based circuits offer scalable pathways for extending this framework to larger systems which we leave open to further research as they remain close to hardware-realistic dynamics.

Looking forward, several promising directions emerge. From a quantum-information perspective, extending this framework to open systems and decohering dynamics would enable entanglement tracking in realistic noisy devices, Generalizing beyond bipartite Rényi-2 entropy to multipartite and alternative entanglement measures (including von Neumann entropy, negativity, and squashed entanglement) would broaden its applicability to quantum networks and many-body architectures. From a machine-learning perspective, incorporating multi-head attention or reinforcement-learning strategies - where randomized-measurement protocols are adaptively optimized to minimize unitary and shot counts - offers a route toward fully autonomous entanglement predittion. Together, these directions position machine-learning-assisted entanglement estimation as a powerful new paradigm for the characterization and control of complex quantum systems as an important open direction.

### ACKNOWLEDGMENTS

A. G. and S. B. acknowledge support from UK Research and Innovation (UKRI) Grant No. EP/R029075/1. R. G. thanks EPSRC grant EP/Y004590/1 MACON-QC for support. D. L. acknowledges support from the EPSRC Centre for Doctoral Training in Delivering Quantum Technologies, grant ref. EP/S021582/1.

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

## A    PRELIMINARIES ON QUANTUM ENTANGLEMENT

As stated in Section 2, we define entanglement as entanglement can be defined as the inseparability of a function in a Hilbert space into two or more constituent subspaces. Let us define an arbitrary composite quantum state as $|\Psi\rangle = |\psi\rangle_A \otimes |\phi\rangle_B$ for Hilbert spaces $\mathcal{H}_A$ and $\mathcal{H}_B$ respectively. If we fix a basis $\{|i\rangle\}_A$ for $\mathcal{H}_A$ and $\{|j\rangle\}_B$ for $\mathcal{H}_B$, then we say a state is entangled if for any vectors $[c_i^A], [c_j^B]$ at least for one pair of coordinates $c_i^A, c_j^B$ we have $c_{ij} \neq c_i^A c_j^B$.

$$|\Psi\rangle = \sum_{i,j} c_{ij} |i\rangle_A \otimes |j\rangle_B \tag{8}$$

Therefore, to encompass all the information stored an entangled state, it is often better to note its ket-bra value, known as its density matrix, $\rho = |\Psi\rangle \langle\Psi|$. By taking the partial trace of this density matrix, we calculate the geometry of our state in the subsystem Hilbert space left after tracing out. In other words,

$$\rho_A := \sum_{j}^{N_B} \left(I_A \otimes \langle j|_B\right) \left(|\Psi\rangle\langle\Psi|\right) \left(I_A \otimes |j\rangle_B\right)$$

The sum occurs over $N_B := \dim(\mathcal{H}_B)$ and $I_A$ the identity operator in $\mathcal{H}_A$. Using this definition, it is trivial to see that the reduced density matrix is a square matrix, and as a result, for a separable state, the product of all non-zero eigenvalues of this reduced density matrix would be 1. However, for an entangled state, the product of the non-zero eigenvalues would be less than 1. Through this lens, we can define our Renyi entropy as a metric using the trace of the reduced density matrix to quantify the entanglement. The Renyi-$\alpha$ entropy of a system, $\rho$, is then

$$H_\alpha(\rho_A) := \frac{1}{1-\alpha} \log_2 \text{Tr}(\rho_A^\alpha) \tag{9}$$

Setting $\alpha = 2$, we recover Equation 1 of the main text.

## B    SIMULATION DETAILS

In Fig 1, we show how we generated our input data. Since we have numerical access to the quantum system through our simulation, we can calculate the entropy (our target value) of the system. However, as also outlined in Section 2, since the Hilbert space dimension scales exponentially with system size, explicitly constructing and diagonalizing the reduced density matrix becomes computationally expensive. So, instead of constructing and diagonalizing the reduced density matrix, we compute the Schmidt decomposition via singular value decomposition of the reshaped wavefunction. This avoids explicitly forming the reduced density matrix and is numerically more efficient in practice. By definition, the Schmidt decomposition states that

$$w = \sum_{i=1}^{m} \alpha_i u_i \otimes v_i \tag{10}$$

where the scalars $\alpha_i$ are real, non-negative, and unique up to re-ordering, for $\{u_1, \ldots, u_m\} \subset \mathcal{H}_1$, $\{v_1, \ldots, v_m\} \subset \mathcal{H}_2$, for any $w \in \mathcal{H}_1 \otimes \mathcal{H}_2$ in Hilbert spaces $\mathcal{H}_1$ and $\mathcal{H}_2$ with dimensions $n$ and $m$ respectively, with Schmidt rank $r < \min(n, m)$.

Comparing this to our definition of a composite state (Equation 8), we define the density matrix as

$$\rho = |\Psi\rangle \langle\Psi|$$

$$= \left(\sum_{i,j} c_{ij} |i\rangle_A \otimes |j\rangle_B\right) \left(\sum_{p,q} c_{pq}^* \langle p|_A \otimes \langle q|_B\right)$$

$$= \sum_{i,j,p,q} c_{ij} c_{pq}^* \left(|i\rangle \langle p|\right)_A \otimes \left(|j\rangle \langle q|\right)_B.$$

where $|i\rangle, |j\rangle, |p\rangle, |q\rangle$ denote the computational basis. To obtain the reduced density matrix of subsystem $A$, we trace over subsystem $B$:

$$\rho_A = \mathrm{Tr}_B(\rho)$$

$$= \sum_{i,j,p,q} c_{ij} c_{pq}^* \left(|i\rangle \langle p|\right)_A \mathrm{Tr}(|j\rangle \langle q|)_B$$

$$= \sum_{i,p,j} c_{ij} c_{pj}^* |i\rangle \langle p|_A = CC^\dagger.$$

If we now perform the singular value decomposition $C = U\Sigma V^\dagger$, then in the Schmidt basis defined by $U$ and $V$, the reduced density matrix becomes

$$\rho_A = U\Sigma^2 U^\dagger.$$

Therefore, its eigenvalues are $\lambda_k = \alpha_k^2$, where $\alpha_k$ are the Schmidt coefficients.

The second Rényi entropy is therefore

$$H_2(\rho_A) = -\log_2 \mathrm{Tr}\left(\rho_A^2\right)$$

$$= -\log_2 \sum_k \lambda_k^2$$

$$= -\log_2 \sum_k \alpha_k^4.$$

where the $\alpha_i$ are our Schmidt eigenvalues.

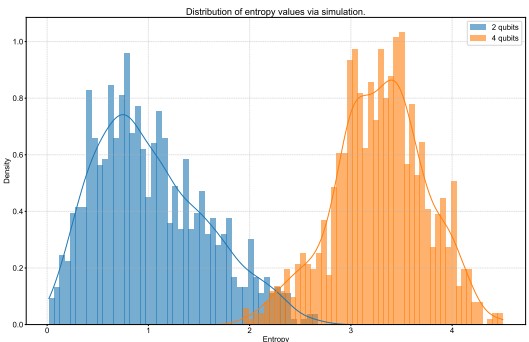

Figure 7: Distribution of entropy values during simulation as a result of Haar randomizing.

It is important to note that in our simulation, we create a Haar random *pure* state. For a chosen bipartition, the entanglement between the two subsystems is quantified by computing the von Neumann entropy of the reduced density matrix obtained by tracing out one half of the system. However, in this sampling method, our distribution of entropy values do not follow a flat, uniform distribution but instead is strongly concentrated around the Page value (Page, 1993) as shown in Fig 7. This concentration is a direct consequence of the geometry of Haar random states and implies that the training targets for our learning model are naturally clustered within a narrow entropy window. While this is appropriate for studying typical properties of Haar states as we have done in this work, the input distribution can be modified to better reflect other physical ensembles when targeting experimentally relevant states with different entanglement structures.

# C   BOUNDING $N_U$ AND $N_M$

Here, we explain on how we approximate the upper bound for our values of $N_M$ and $N_U$. A single measurement produces one classical outcome sampled from the probabilities and numerically this is encoded as a one-hot value. The probabilities used for sampling are the diagonal elements of the density matrix in the computational basis $\langle i^{\otimes n}| \rho |i^{\otimes n}\rangle$. These diagonal elements are approximated by repeated measurements of the density matrix. In the left plot of Fig 8, we plot out the deviation of the measurement to the actual diagonal values and observe that the number of qubits only very weakly influences the number of measurements needed to estimate the diagonal values of the density matrix.

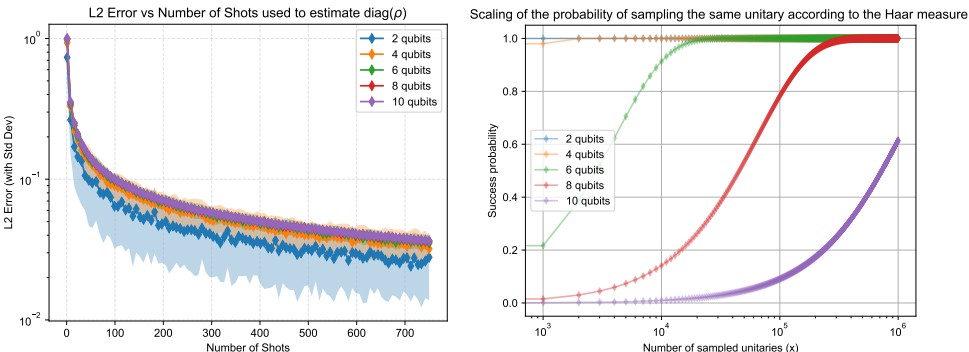

Figure 8: *Left:* Number of measurements (shots) per quantum system used to determine the diag($\rho$). Each qubit size is simulated 100 times to get the L2 error. *Right:* Scaling of number of unitaries needed to be sampled for different number of qubits.

However, only the diagonal values of the density matrix do not give us enough information about the entanglement of the system. By rotating our system, by the unitaries discussed above, we can probe different combinations of the elements of the density matrix revealing off diagonal elements. Suppose we have a density matrix of $n$ qubits, $\rho$, and a unitary to rotate our system, $U \in SU(2^n)$, then the diagonal elements of $\rho$ transform as,

$$\rho'_{mn} = U_{mj}\rho_{jk}U^{\dagger}_{kn}$$

where we opt to use the Einstein summation, summing over indices $j, k$. Since we can only measure the diagonals and by assuming we can perform $N_M$ measurements to estimate our diagonal, our transformation reduces to,

$$\rho'_{dd} = U_{dj}\rho_{jk}U^{\dagger}_{kd}$$

and for any arbitrary row/column of our unitary,

$$\rho'_{dd} = \vec{U}^{(d)}_j \rho_{jk} (\vec{U}^{\dagger}_k)^{(d)}$$

where we use $(d)$ to indicate the row/column. Since $\rho_{jk}$ does not depend on $d$, we see that our rotated density matrix elements represent a linear combination of all matrix elements of $\rho$. Therefore, the information offered by a single transformation is independent of other unitaries and insufficient to determine our original density matrix. The question then arises of how many unitaries do we need to probe and recover the off diagonal elements of our original density matrix.

It is well known that the probability of sampling the same unitary on the Haar measure of $n$ qubits scales as $4^n$ Schuster et al. (2025). We plot these probabilities in Fig 8 (right), where we notice the complexity in scaling to a higher number of qubits - namely, for 10 qubits, we need an upwards of $10^6$ unitaries to sample the same unitary twice, thereby encompassing the entire space. This scaling is also one of the limitations of this work, as discussed in Section 5.

# D  TRAINING AND EVALUATION DETAILS

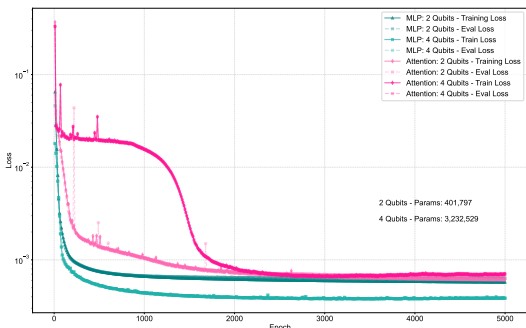

Figure 9: Loss curves of both models. Hot pink corresponds to the self-attention model while the blue curve corresponds to the feed forward MLP. Lowest validation loss models are stored for the best models. Validation done every 10 epochs.

## D.1  TRAINING CONFIGURATION

Our training configuration is summarized in Table 1.

Table 1: Training hyperparameters. Model architecture is described in the main text.

| Hyperparameter | Value |
|---|---|
| Optimizer | Adam |
| Learning rate (Attention) | $5 \times 10^{-8}$ |
| Learning rate (MLP) | $1 \times 10^{-5}$ |
| Batch size | 32 |
| Number of epochs | 5000 |
| Loss function | L2-loss |

## D.2  COMPUTE ENVIRONMENT

Our compute details is summarized in Table 2.

Table 2: Compute and software environment.

| Component | Details |
|---|---|
| GPU | NVIDIA Tesla V100 (32 GB) |
| Number of GPUs | 4 |
| RAM | 512 GB |
| Framework | Jax 0.4.28 |
| Training time (2 qubits, MLP) | $\sim 10$ hours |
| Training time (2 qubits, Attention) | $\sim 10$ hours |
| Training time (4 qubits, MLP) | $\sim 46$ hours |
| Training time (4 qubits, Attention) | $\sim 50$ hours |

## D.3  MODEL DETAILS

As mentioned there are two neural networks - one for processing the unitary $\phi$ and another for processing the outcome probabilities and the processed unitary together $S$. Both these models had

the same number of parameters per qubit number - the 2 qubit model had $401,797$ parameters while the 4 qubit model had $3,232,529$ parameters. For 2 qubits, $\phi$ had hidden dimensions $[128, 512, 128]$ and $S$ had $[128, 512, 128]$. For 4 qubits, $\phi$ had $[1024, 512, 128]$ and $S$ had $[128, 512, 1024, 512, 128]$. Further, residual connections with layer normalization were implemented with no skip connections. All networks had ReLU() activation. Checkpoints saved every epoch and the best model selected by highest validation accuracy.

# E  ADDITIONAL FIGURES

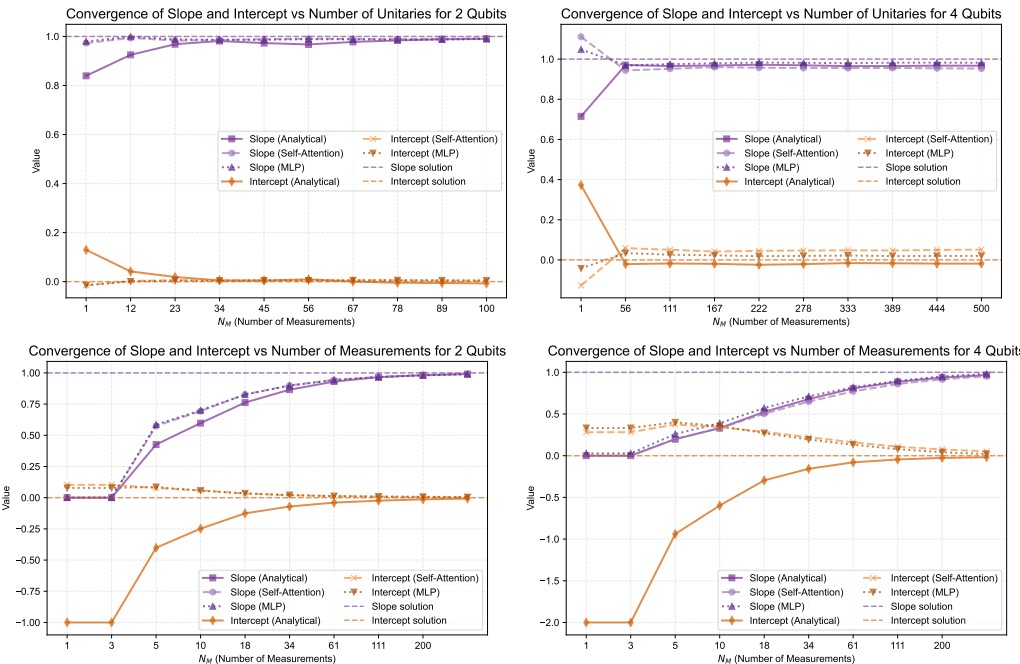

Figure 10: Progression of slope and intercept in the line of best fit of Fig 3 and Fig 5. In the figures of the main text, the scatter plots depict the true against predicted value; by considering the line of best fit, the ideal model would have slope 1 and intercept 0. We plot these values for different $N_U$ values in the bottom two plots and for different $N_U$ values in the bottom plot, for 2 and 4 qubits.

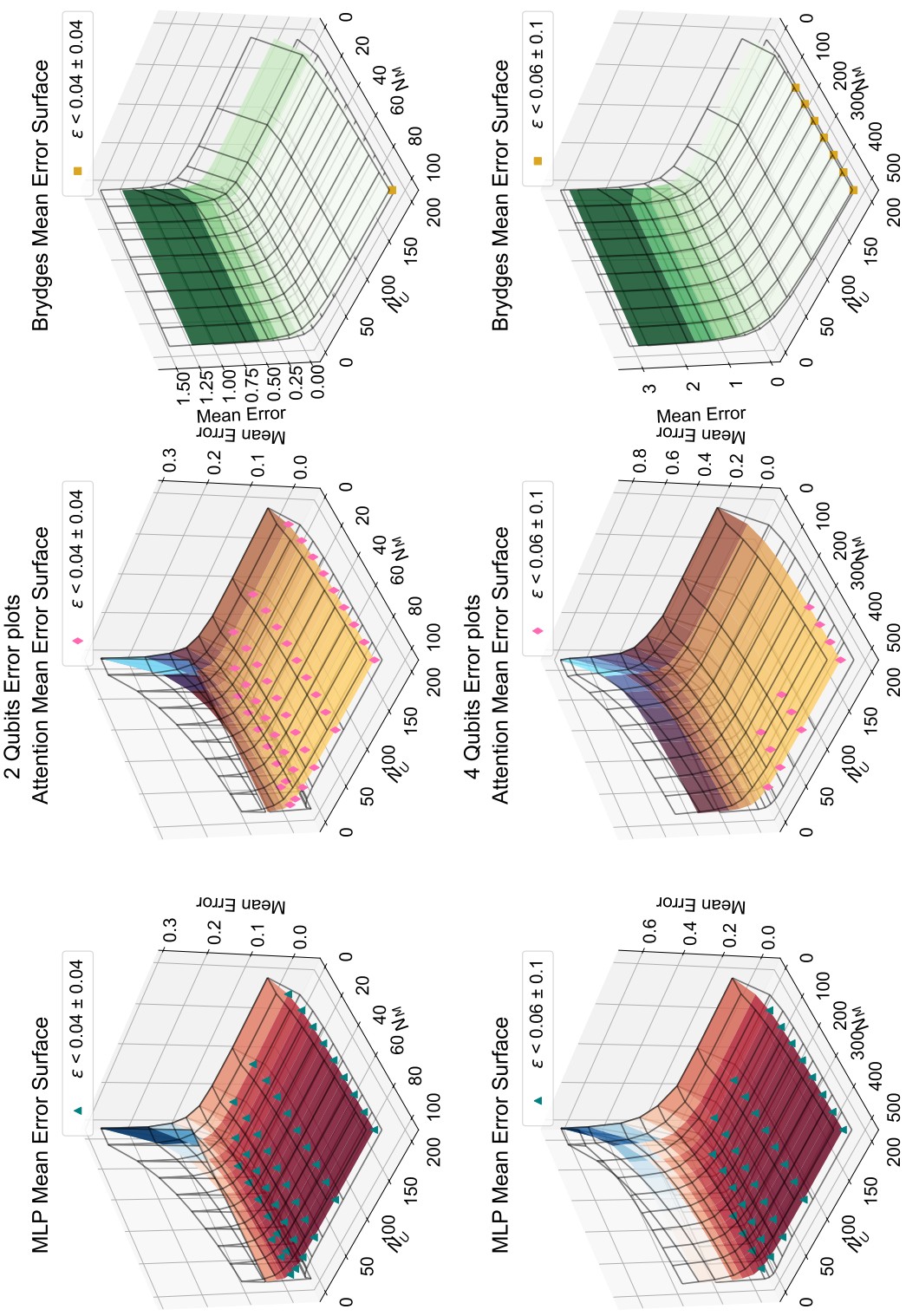

Figure 11: 3D extrusions of Fig 4 and Fig 6. We extrude the figures by plotting the error on the z-axis. Note that the wireframe correspond to the variance in the error.

