# OpenReview forum: "Self-Attention for Quantum Entanglement Prediction"
_ICLR.cc/2026/Workshop/FM4Science — ICLR 2026 Workshop FM4Science Poster_

### Official Review · Reviewer_1Wct · 2026-02-19
**Interesting Application but Weak Alignment with Foundation Model Theme**

**Rating:** 4
**Confidence:** 4

**Review:**

The authors compare an analytical estimator with a feed-forward neural network (MLP) and a self-attention model. Experiments are conducted on 2- and 4-qubit systems, analyzing the impact of different (N_U, N_M) combinations on estimation accuracy. The authors claim that machine learning models reduce the required number of measurement shots and random unitaries, thereby enabling more efficient entanglement estimation. The paper presents a technically sound and well-structured study with a meaningful scientific motivation. However, the core model does not demonstrate a clear advantage, the experimental scale is limited, and the connection to foundation models is weak.

Strengths

The theoretical background is well presented, with clear explanations of classical shadows, and the relevant formulas. The research problem is scientifically meaningful, as efficient entanglement quantification under limited measurement resources is an important challenge in quantum information science. The experimental setup is systematic, and the roles of N_U (number of unitaries) and N_M (number of measurements per unitary) are carefully analyzed. Overall, the structure of the paper is coherent and logically organized.

Weakness

The workshop theme is “Foundation Models for Science: Real-World Impact and Science-First Design.” However, the paper mainly focuses on comparing three models at a limited system scale. It does not involve foundation model, pretraining, large-scale modeling, scaling laws, or cross-system generalization. The overall contribution is closer to a small-scale supervised learning application rather than a foundation-level scientific modeling effort. In addition, the paper does not demonstrate generalization across systems or tasks, nor does it include validation on real quantum hardware or noisy experimental data, limiting its connection to “Real-World Impact.”


Although the title emphasizes self-attention, the experimental results show that the MLP consistently achieves comparable or better accuracy with fewer resources. The self-attention model does not demonstrate a clear or consistent advantage over the simpler baseline, which weakens the central contribution implied by the title.

All experiments are restricted to 2- and 4-qubit systems. At this scale, the Hilbert space dimension remains relatively small, and classical simulation is tractable. Therefore, claims regarding scalability and performance persistence with increasing system size are not empirically substantiated.

---

### Official Review · Reviewer_URJh · 2026-02-24
**Promising low-shot results but more clarification needed for attention and scaling claims**

**Rating:** 7
**Confidence:** 3

**Review:**

The experimental protocol is reasonably systematic over $(N_U,N_M)$ grids and includes a clear analytical baseline and two learned models. However, the scope is limited to simulated Haar-random pure states and only 2 and 4 qubits for the entanglement-prediction task are reported. Some broad claims like those about range of qubit numbers and scaling appear stronger than evidence presented. Definitions and derivations are complete and comprehensive, though some wording/notation like "best model selected by highest validation accuracy" may be confusing. The paper states MLP can be more resource-efficient than attention in some regimes, but clearer ablation or conceptual argument that self-attention provides a unique advantage over MLP on the shown tasks strengthens its originality. The demonstrated setting is narrow and currently limits the strength of the significance claim for "large-scale quantum devices". If the reported reduction in required shots/measurements holds broadly, it could be practically relevant for near-term experiments. In addition to the next steps in the paper, potentially report minimal $(N_U,N_M)$ required to hit a fixed target MSE with confidence, and add a simple synthetic noise model beyond shot noise to more directly support the NISQ framing.

Overall, this paper has a well-scoped problem formulation with explicit estimators. That said, this paper can improve on the following:
i) The data distribution is Haar-biased toward near-maximal entanglement (Fig. 7), but robustness to other ensembles, low-entanglement regimes, or realistic device noise is not empirically tested.
ii) Baselines and ablations are limited to Brydges vs the two networks, and the reported compute cost (Table 2) is not analyzed against the claimed measurement savings.

Last but not least, I have the following questions:
- How exactly are the attention weights applied at inference when $QK^\top$ is "frozen"? If submitted to other venues, maybe consider providing explicit tensor shapes and a step-by-step forward pass.
- In Eq. (7), is $A$ intended as a $(N_U\times N_U)$ attention matrix over unitaries, or a feature-space covariance/projector? The current definition $A=\sum_s|u^{(s)}\rangle\langle u^{(s)}|$ suggests the latter; please clarify how it interfaces with Eq. (5).
- The paper claims generalization “to other forms of entanglement”. What specific measures are intended and could at least one experiment be included to demonstrate this?

---

### Official Review · Reviewer_yDwb · 2026-02-26
**This paper develops a feed-forward neural network and an attention-based network for quantum entanglement prediction.**

**Rating:** 5
**Confidence:** 2

**Review:**

This paper develops a feed-forward neural network and an attention-based network for quantum entanglement prediction. This paper describes the architecture of the two models for predicting quantum entanglement and performed benchmarking experiments to compare this deep learning-based approach with conventional approaches to highlight the effectiveness of the proposed model.

My major concern is that this work does not involve the use of foundational models, albeit it is tightly related to the application of deep learning for science. This focus might be unrelated to the focus of this workshop.

---

### Official Review · Reviewer_q9dy · 2026-02-26
**This paper studies the problem of estimating quantum entanglement from limited measurement data. The authors propose two machine learning models- a feed-forward neural network and a self-attention–based architecture—to predict the Rényi-2 entanglement entropy of Haar-random pure quantum states using classical shadow measurements. The approach avoids full state tomography and is claimed to be scalable, permutation invariant, and suitable for near-term quantum devices. Empirical results suggest that the learning-based models outperform conventional analytical estimators across multiple system sizes, with the attention-based model achieving the best performance.**

**Rating:** 7
**Confidence:** 2

**Review:**

Strengths

- The paper addresses a unique application area- namely efficient entanglement quantification from experimentally accessible data.

- The use of classical shadows as input aligns well with current experimental protocols and avoids unrealistic assumptions.

- The motivation for using self-attention architectures is clear, particularly with respect to permutation invariance and scalability.

- The method has potential practical impact for near-term quantum devices, as it avoids full quantum state tomography and wave-function reconstruction.

-The paper is generally well written. The introduction provides sufficient background and context, and the relationship to prior work is discussed. However, additional details on training procedures, model scaling, and experimental assumptions would improve clarity.

Weaknesses

- The conceptual novelty is somewhat limited, as the machine learning architectures employed are standard and the contribution lies primarily in their application to this task.

- The evaluation is restricted to Haar-random pure states, which are mathematically convenient but not representative of many physically relevant quantum states.

- The baselines appear limited to analytical estimators, with no comparison to other learning-based or permutation-invariant methods.

- The paper provides limited insight into what the attention mechanism is learning, and interpretability is not addressed. Robustness to realistic experimental noise (e.g., shot noise or measurement errors) is not evaluated.

Questions for the Authors

- How well does the proposed approach generalize to non-Haar ensembles, such as states generated by shallow random circuits or noisy quantum processes?

Overall Recommendation

Recommendation: Weak Accept / Borderline
Confidence: Medium

---

### Meta-Review · Area_Chair_kDEu · 2026-02-27

**Recommendation:** Accept (Poster)
**Confidence:** 2

**Metareview:**

The work addresses a scientifically meaningful problem—efficient entanglement quantification under realistic measurement constraints—and seems to be well written and technically sound. In particular, it shows a nice connection to classical shadow protocols and potential near-term practical relevance. However, across reviews, concerns emerge regarding limited conceptual novelty (standard architectures applied to a new task), narrow experimental scope (Haar-random states, 2–4 qubits only), insufficient baselines and ablations, lack of robustness tests under realistic noise, and unsubstantiated scaling or “large-device” claims. Overall it is a solid core application and aligns with one of the workshop's themes: scientific ML.

---

### Decision · Program_Chairs · 2026-03-03

Accept (Poster)